# The Effect of MICP on Physical and Mechanical Properties of Silt with Different Fine Particle Content and Pore Ratio

Yang Zhao [1], Qian Wang [1], Mengnan Yuan [1], Xi Chen [1], Zhiyang Xiao [1], Xiaohong Hao [1], Jing Zhang [2,*] and Qiang Tang [3,*]

1. College of Geosciences and Engineering, North China University of Water Resources and Electric Power, Zhengzhou 450046, China; china.zhaoyang@hotmail.com (Y.Z.); wq13276901507@163.com (Q.W.); ymengnan1223@163.com (M.Y.); cx1055216675@163.com (X.C.); 201401519@stu.ncwu.edu.cn (Z.X.); haoxiaohong@ncwu.edu.cn (X.H.)
2. Department of Neuroimmunology, Henan Institute of Medical and Pharmaceutical Sciences, Zhengzhou University, Zhengzhou 450052, China
3. School of Rail Transportation, Soochow University, Suzhou 215131, China
* Correspondence: sgrzdy@163.com (J.Z.); tangqiang@suda.edu.cn (Q.T.)

**Abstract:** Microbial-induced calcium carbonate precipitation (MICP) is a new soil remediation technology, which can improve the physical and mechanical properties of soil by transporting bacterial solution and cementation solution to loose soil and precipitating calcium carbonate precipitation between soil particles through microbial mineralization. Based on this technique, the effects of different fine particle content and pore ratio on the physical and chemical properties of silt after reinforcement were studied. The content of calcium carbonate, the ability of silt to fixed bacteria, unconfined compressive strength (UCS), permeability coefficient and microstructure of the samples were determined. The results showed the following: In the process of calcium carbonate precipitation induced by microorganisms, more than 50% bacterial suspension remained on the surface of silt particles and their pores. The higher the bacterial fixation rate of silt, the more $CaCO_3$ was generated during the solidification process. The bacterial fixation rate and $CaCO_3$ content both decreased with the increase in the pore ratio and increased with the increase in the fine particle content. XRD and SEM images show that the calcium carbonate is mainly composed of spherical vaterite and acicular cluster aragonite. There is an obvious correlation between unconfined compressive strength and $CaCO_3$ content of silt. When $CaCO_3$ content accumulates to a certain extent, its strength will be significantly improved. The unconfined compressive strength of silt A with pore ratio of 0.75 and fine particle content of 75% is 2.22 MPa when the single injection amount of cementing fluid is 300 mL. The permeability coefficient of cured silt can be reduced by 1 to 4 orders of magnitude compared with that of untreated silt. In particular, the permeability of MICP-treated silt A is almost impermeable.

**Keywords:** calcium carbonate precipitation induced by microorganisms; silt; void ratio; fine particle content; unconfined compressive strength; permeability coefficient; XRD; SEM

## 1. Introduction

Microbial-induced calcite precipitation (MICP) is a bacteria-induced bio-mineralization process, which has been paid close attention to civil, infrastructure and environmental engineering [1–3]. Taking advantage of natural biological processes, it contributes to additional cementation at particle to particle contacts of soils, i.e., strength, stiffness or permeability improvement of soils [4–6]. In a more environmentally friendly and sustainable manner, calcium carbonate precipitation induced by microorganisms is a promising solution for soil-related engineering problems.

To date, most studies of microbial-induced calcium carbonate precipitation have focused on treating various kinds of sands [7–9]. Few studies have been done on other soil types. However, Dejong et al. [10] and Mortensen et al. [11] considered calcium carbonate precipitation induced by microorganisms has the potential to improve silt. Tropical residual

soil classified as silt was tested by Soon et al. [12,13], and the greatest improvements in shear strength and reduction in hydraulic conductivity achieved are 100% and 90%, respectively. Zamani et al. [14] studied the changes of hydraulic conductivity of fine sand and silt after MICP treatment, and the test proved that the application of MICP reduced the hydraulic conductivity of soil, while improving its strength and stiffness. By adding finer particles changing the grain size distribution, Jiang et al. [15] concluded samples treated by microorganisms had better fine particle content behavior better in the erosion test. Li et al. [16] indicates that incorporation of fly ash in Biological cement reaction is an effective means of increasing the strength of soils. Moreover, void ratio (relative density) is taken into consideration on the improvement of soils. Gao et al. [17] conducted triaxial consolidation and drainage tests on sand with different relative compactness (Dr = 30%, 50%, 90%), and found that the strength and deformation control of biocemented loose and moderately compacted sand could reach or exceed that of compacted sand.

However, experimental data on microbial cemented silt are still very limited considering the variation of both grain size distribution and void ratio. Therefore, the effects of grain size distribution and void ratio on the bio-mineralization process and mechanical properties of microbial cemented silt require further investigation. In the present study, three different types of silt samples with two different void ratios were treated with biological cementation, and the yield of calcium precipitation and uniaxial compression strength (UCS) of the silt samples after biological cementation were evaluated and discussed.

## 2. Materials and Methods

### 2.1. Biological Treatment Process

In the study, urease-active strain Sporosarcina pasteurii (CGMCC 1.3687) was used, provided by the China General Microbiological Culture Collection. It is an aerobic bacterium with a diameter of 2 μm to 3 μm. Optical density was measured using a spectrophotometer at a length of 600 nm ($OD_{600}$), and urease activity was measured immediately after sampling at 25 °C by the conductivity method [18].

Under aerobic batch conditions at 32 °C, bacteria were cultivated in a medium containing 5 g/L soy peptone, 3 g/L beef extract, 0.02 g/L $NiCl_2$, and 20 g/L $CO(NH_2)_2$, at a pH of 8, to late exponential phase with $OD_{600}$ of $3 \pm 0.5$ and urease activity of $18 \pm 2$ mM urea/min.

### 2.2. Soil Properties and Gradation

The soil used in the experiment was taken from the floodplain area of the middle and lower reaches of the Yellow River (Zhengzhou and Jinan, China), as shown in Figure 1. According to the particle size larger than 0.075 mm, using the sieving method, and less than 0.075 mm using the hydrometer method to carry out particle separation test on the soil in the two areas, the particle grading curve of the two is shown in Figure 2, in which the fine particle content of the soil in Zhengzhou area is 75%, and the fine particle content of the soil in Jinan area is 47%. In order to obtain the soil with fine particle content between the two areas, the Zhengzhou soil and Jinan soil were mixed in a ratio of 3:7, and the fine particle content of the mixed silt was 63%. As can be seen from the specification, the soil whose particle size is less than 0.075 mm and the mass of the fine-grained group is greater than or equal to 50% of the total mass in the sample is called fine-grained soil (GBJ 145-90), so the three kinds of soil are all fine-grained soil. The liquid limit, plastic limit and plastic index of the three kinds of soil were measured by the combined liquid plastic limit instrument, as shown in Table 1. The three kinds of fine-grained soil were further subdivided according to the plastic diagram (liquid limit 17 mm), and all of them were low-liquid limit silt soil.

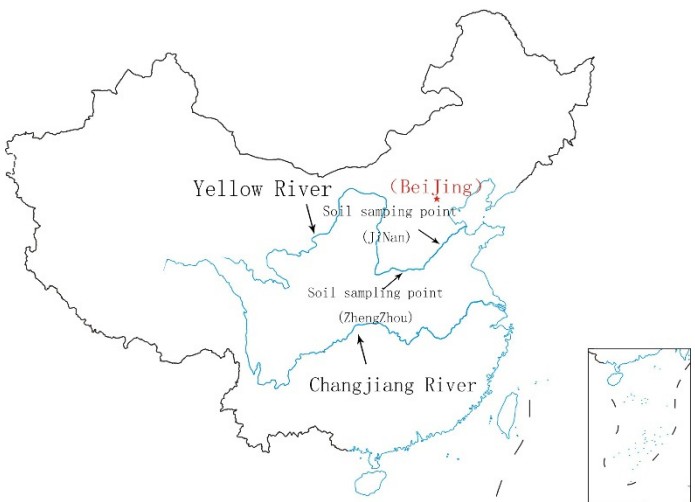

**Figure 1.** The location of the test soil.

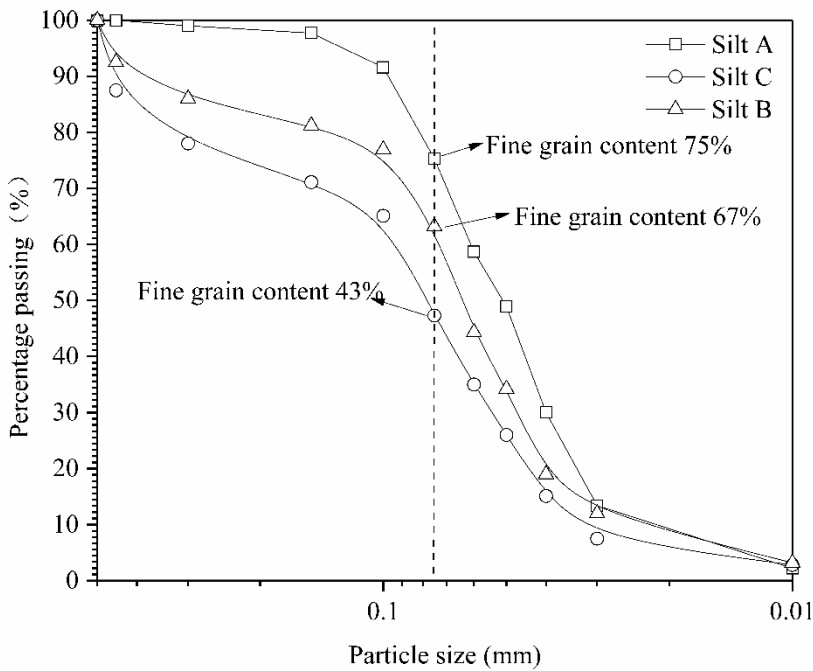

**Figure 2.** Grading curve of soil particles.

**Table 1.** Physical characteristics of soil.

| Soil Type | Physical Properties | Liquid Limit (%) | Plastic Limit (%) | The Plastic Index |
|---|---|---|---|---|
| Silt A | | 30.5 | 14.2 | 16.3 |
| Silt B | | 24.8 | 10.9 | 13.9 |
| Silt C | | 24.5 | 11.3 | 13.2 |

Two porosity ratios (0.75 and 0.9) were considered, and a plexiglass mold with an inner diameter of 30 mm and a length of 60 mm was used. All samples were prepared by static compaction. For each type of silt sample, control the weight of the soil to achieve the desired two porosity ratios. Six types of samples were prepared by considering three kinds of fine particle contents and two kinds of pore ratios. For the convenience of description, Silt A, Silt B and Silt C are used to represent the three silts according to their fine particle

contents from high to low, respectively. The pore ratios of 0.75 and 0.9 are used to represent I and II, respectively.

### 2.3. MICP Treatment

A two-phase injection scheme with peristaltic pump was used at $25 \pm 2\,^{\circ}\text{C}$ [19]. From top to bottom, bacterial suspensions of 30 mL were injected into the sample at 1.0 mL/min, and $OD_{600}$ of the effluent was tested after injecting. Then, 150 mL cementation solutions of the urea and $Ca(CH_3COO)_2$ with 1 M equalmolar concentrations at 1.5 mL/min were pumped. After a curing time of 8 h, cementation treatment was repeated once for each individual sample. In order to study the effect of the injection amount of cement on the physical and chemical properties of silt, the injection amount of cement was increased to 300 mL per round as a control when other factors remained unchanged. $Ca(CH_3COO)_2$ was adopted here following by Zhang et al. [20], who considered $Ca(CH_3COO)_2$ is a better source of calcium compared with $CaCl_2$ and $Ca(NO_3)_2$. The grouting device is shown in Figure 3. After the end of grouting, the treated samples were static and dried, and then measured for various physical and chemical properties. Each test was carried out in three groups of parallel tests. In order to better evaluate the influence of different factors on the physical and chemical properties of silt, additional blank control samples were prepared.

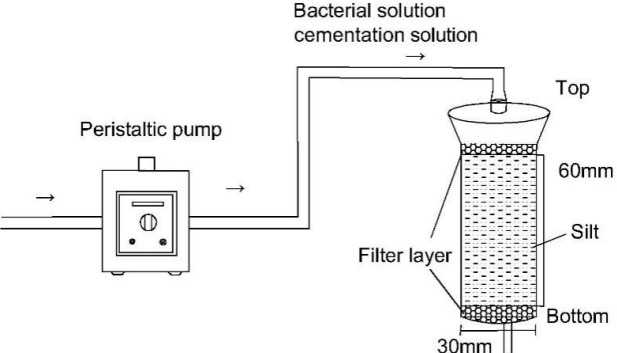

**Figure 3.** Grouting device.

### 2.4. Tests and Methods

Samples cured by microorganisms were immersed in deionized water for 48 h to eliminate the soluble substances on the surface, and then the biological treated and control samples were dried at $60\,^{\circ}\text{C}$ for 48 h. The corresponding dry mass was denoted as $M_d$. Unconfined compressive strength tests were carried out for all the samples at a loading rate of 1 mm/min. After the unconfined compressive strength tests, all fractions of each biological treated sample were collected and mixed with excessive 2 M HCl, and then the final residues were collected by filter paper and were dried at $105\,^{\circ}\text{C}$ for 24 h. The dry mass of the soil was subsequently measured and denoted as $M_{silt}$. Hence, the mass of calcium carbonate $M_{Ca}$ was calculated through the result of the sample before and after treatment [12,21]:

$$M_{Ca} = M_d - M_{silt} \tag{1}$$

The calcium carbonate content, $C_m$, was defined as the ratio of the dry mass of calcium carbonate crystals and mixed sands, which can be expressed as:

$$C_m\ (\%) = M_{ca}/M_{silt} \times 100\% \tag{2}$$

The enzyme activity of the bacterial suspension was measured as $U_{ini}$ before grouting, and the enzyme activity of the effluent bacterial suspension was measured as $U_{eff}$ after the bacterial suspension was injected into each sample. The change in bacterial activity can be

assessed by the difference in urease activity between the injected bacterial suspension ($U_{ini}$) and the effluent suspension ($U_{eff}$) and determined by the following equation [19]:

$$Bacterial\ activity\ retention\ (\%) = (1 - U_{eff}/U_{ini}) \times 100\% \qquad (3)$$

It can be seen that the more retained bacterial cells and urease released in the samples, the higher the retention of bacterial activity.

Mineralogical composition was examined by X-ray diffraction (XRD) test, and microstructure was examined by the mercury intrusion porosimeter (MIP) tests and scanning electron microscope (SEM) tests. For this MIP and SEM test, trimmed small pieces were all from the central of each sample.

## 3. Results and Discussion

### 3.1. Generated Calcium Carbonate

As can be seen from Figures 4 and 5, after curing by microorganism, the content of calcium carbonate in silt is between 9% and 20%, and the generated calcium carbonate will attach to the surface of silt particles or fill in the pores between particles, cementing the loose silt particles into columns with a certain strength. This was confirmed by Montoya et al. [22], it is believed that precipitated calcium carbonate influences soil behavior by bonding particles together and densifying the soil by pore space precipitation, which decreases its void ratio. When the injection amount of cementing fluid was increased to 300 mL each round, the production amount of $CaCO_3$ was still significantly increased by controlling other factors unchanged. This indicated that the production amount of $CaCO_3$ could be increased by appropriately increasing the injection amount of cementing fluid. However, the content of calcium carbonate does not increase linearly with the amount of cement injection. On the one hand, the urease activity of the bacterial fluid is limited, which cannot promote the hydrolysis of more urea to produce $CO_3^{2-}$ and $Ca^{2+}$ binding. On the other hand, with the injection of bacterial liquid and cementing liquid, $CaCO_3$ is constantly generated in the pore between particles, which makes the silt structure more compact, and it is difficult to inject subsequent bacterial liquid and cementing liquid.

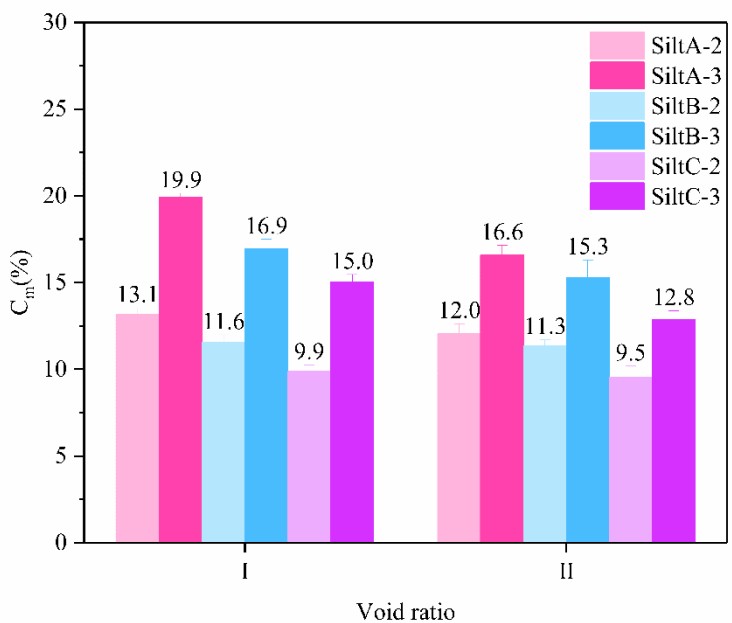

**Figure 4.** Effect of different pore ratio on calcium carbonate production.

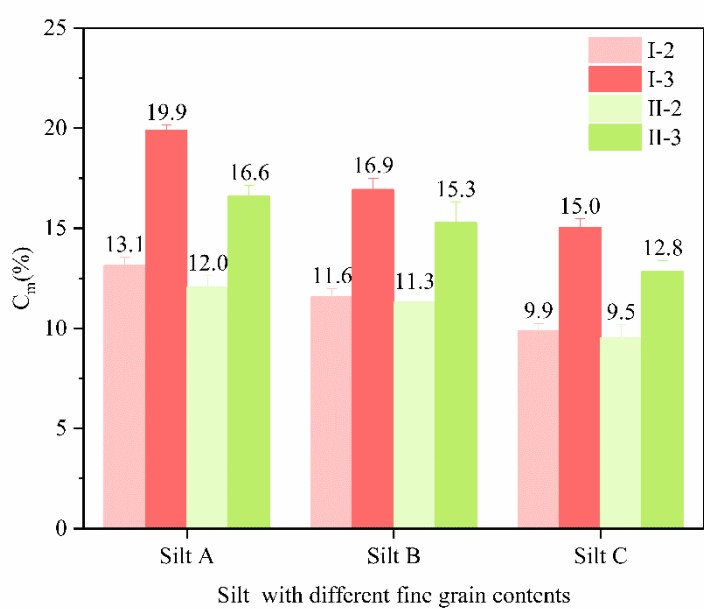

**Figure 5.** Effect of different fine particle content on calcium carbonate production.

As can be seen from Figure 4, when the content of fine particles is the same, more $CaCO_3$ is generated in silt with a pore ratio of 0.75. This is because the silt sample with a pore ratio of 0.9 has a loose structure and a large proportion of large pores. In the process of grouting, bacterial liquid and cementing liquid are more likely to flow out under the action of peristaltic pump, which is not conducive to the generation of $CaCO_3$. As can be seen from Figure 5, when the porosity ratio is the same, $CaCO_3$ production increases with the increase in fine particles content, fine particles will increase the adhesion rate of bacterial fluid, and $CaCO_3$ is more easily generated on the surface of particles. It can be seen that after curing by microorganism, the maximum $CaCO_3$ production amount is produced in silt A with pore ratio of 0.75 and fine particle content of 75%. Md Touhidul Islam et al. [23] believed that this increase in calcium carbonate precipitation with the increase in clay content indicates that the activity of soil bacteria present in the clay portion of the soil was essential for precipitating calcite.

*3.2. SEM/XRD*

Calcium carbonate generated from biological treated samples Silt A, Silt B and Silt C was analyzed by XRD, as shown in Figure 6a–c, respectively. Three polymorphs types of calcium carbonate, including calcite, vaterite and aragonite, were identified in three samples by comparison with standard cards. In sample A, the calcium precipitation consisted of 81.7% vaterite, 17.5 aragonite, and 0.8% calcite (Figure 6a). In sample B, the calcium precipitation consisted of 73.8 vaterite, 20.6% aragonite, and 5.6% calcite (Figure 6b) In sample C, the calcium precipitation consists of 60.3% vaterite, 32.3 aragonite, and 7.4% calcite (Figure 6c). This indicates that the calcium polymorph types of the three samples are mainly composed of vaterite and aragonite, and the calcium precipitation polymorph type composition in the samples is not affected by the content of fine particles. Calcite, vaterite and aragonite are the three polymorph types of calcium carbonate, and many researchers have found that calcite is the main calcium polymorph type in the process [24]. There may be two reasons for this change. First, spherical arponite is the primary mineral formed at high levels of hydrolysis (above 18 mM Urea h$^{-1}$) [25]. In addition, calcium sources influence the polymorph type of calcium [20]. Calcite may be formed when $CaCl_2$ is used, while vaterite and aragonite precipitate easily when $Ca(CH_3COO)_2$ is used. This result is consistent with the research results of using $Ca(CH_3COO)_2$ as calcium source, and the main calcium polymorph types are vaterite and aragonite [20].

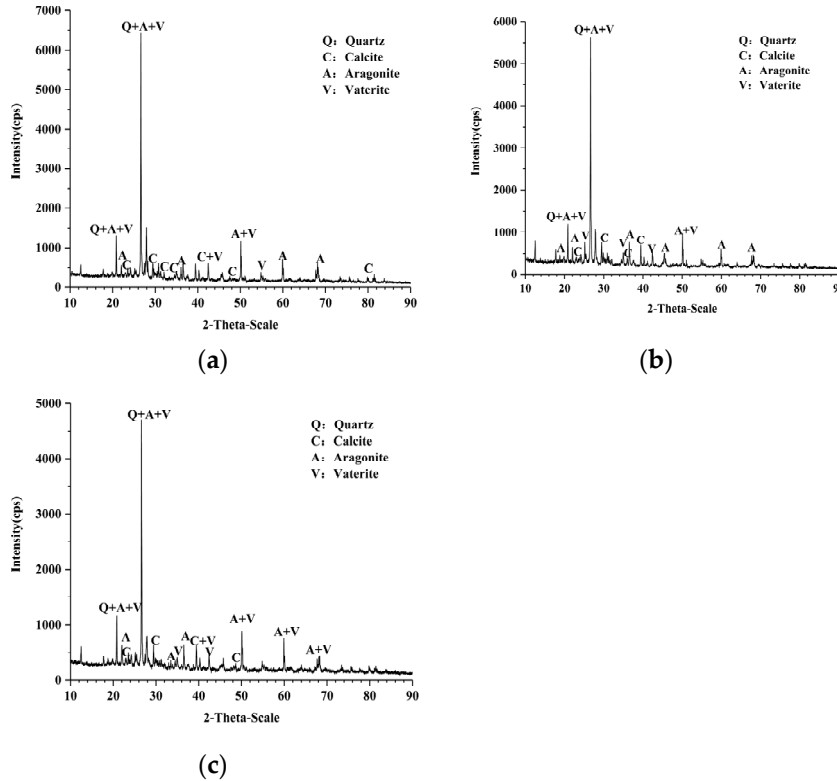

**Figure 6.** X-ray diffraction results of different silts. (**a**) X-Ray diffraction results of Silt A; (**b**) X-Ray diffraction results of Silt B; (**c**) X-Ray diffraction results of Silt C.

At room temperature, calcite is the most stable crystalline phase, while needle-like aragonite and spheroidal aragonite are metastable crystalline phases, which can be transformed into stable crystalline calcite under certain conditions. Different from calcite and needle-like aragonite, spheroidal aragonite has more excellent water solubility, spherical and porous structure and high specific surface product. Few literatures have explored the relationship between the crystal phase of $CaCO_3$ and experimental conditions in the presence of ammonium salts in a gas–liquid system. Studies have shown that under the condition of pH = 11.1, spheraragonite and calcite are generated at the same time, but with the decrease in pH value, the composition of calcite decreases, until pH = 7.9 only spheraragonite crystal phase.

SEM images of three different fine particle contents of silty soils treated with calcium carbonate precipitation induced by microorganisms were obtained to examine the morphology of samples. After curing by microorganism, obvious mineral precipitation traces can be observed on the surface of silt particles and between pores (Figure 7a–c). At the same time, two different shapes of mineral polymorphs (spherical vaterite and acicular cluster aragonite) can be identified (Figure 7b,c), which are consistent with the corresponding mineral composition revealed by XRD tests. Compared with samples B and C, the calcium polymorph type of sample A is mainly spherical vaterite, and more calcium carbonate polymorphs are deposited on its surface with more uniform size (Figure 7a). With the decrease in fine particle content in silt, the content of acicular cluster aragonite in samples B and C increases (Figure 7b,c). Chu et al. [26] believed that the biological blocking and biocementation of sand particles caused by calcareous precipitation enhanced the strength of sand particles, in other words, a more obvious effect led to a more compact structure.

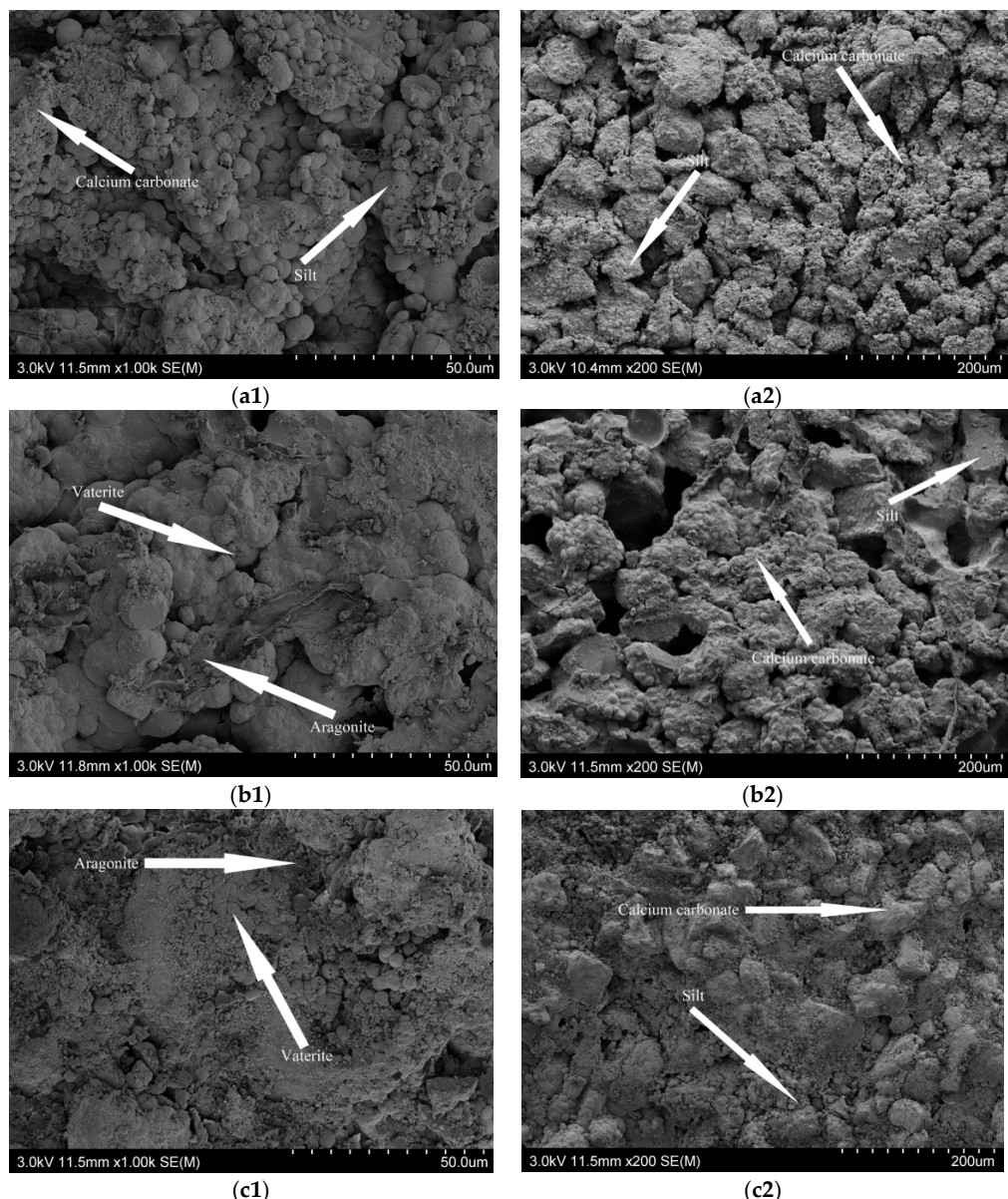

**Figure 7.** SEM images of different silts. (**a1**,**a2**) SEM image of silt A; (**b1**,**b2**) SEM image of silt B; (**c1**,**c2**) SEM image of silt C.

### 3.3. Ability to Retain Bacteria

As can be seen from Figure 8, when the pore ratio was 0.9, the capacity of residual bacteria decreased by about 10% compared with that of the same type when the pore ratio was 0.75. When the pore ratio was 0.75, the retention capacity of bacteria was between 60% and 80%, which indicated that more than half of the bacteria remained in the silt during the first injection of bacterial liquid. The higher the retention capacity of bacteria, the more Bacillus pasteurii octahedrin used for catalyzing urea hydrolysis. In the process of grouting, the bacteria liquid is usually injected first and then the cement liquid is injected. For the controls, similar to the findings of Harkes et al. [19], more than 70% of bacterial activity could not be retained. The retention capacity of bacteria is related to the bacterial suspension $OD_{600}$ that flows out during the first round of grouting, and has nothing to do with the injection amount of cement liquid. Therefore, there is little difference between $OD_{600}$ when the injection amount of cement liquid is 150 mL and 300 mL. As can be seen from Figure 9, in the case of the same pore ratio, the capacity of silt retention bacteria increases with the increase in fine particle content, which is also related to the adsorption

of fine particles. It can be seen from the figure that silt A with pore ratio of 0.75 and fine particle content of 75% has the highest bacteria retention capacity, which is 77%.

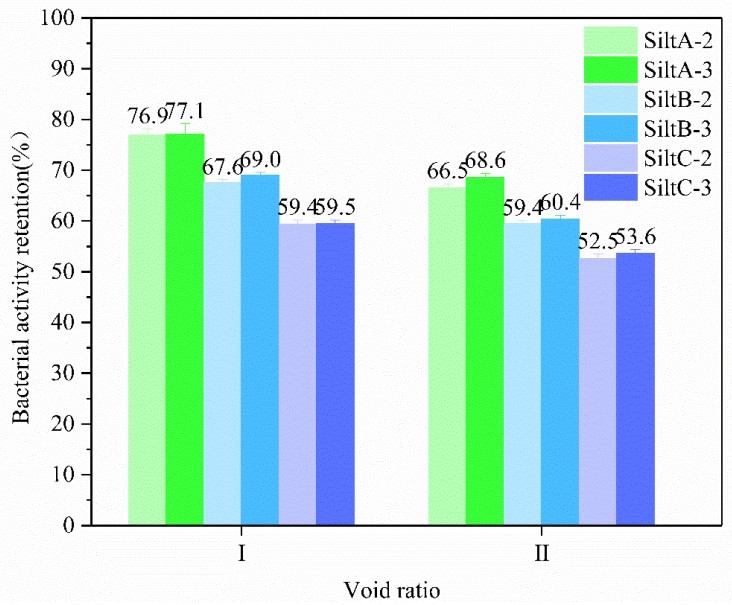

**Figure 8.** Effect of different pore ratios on the ability to retain bacteria.

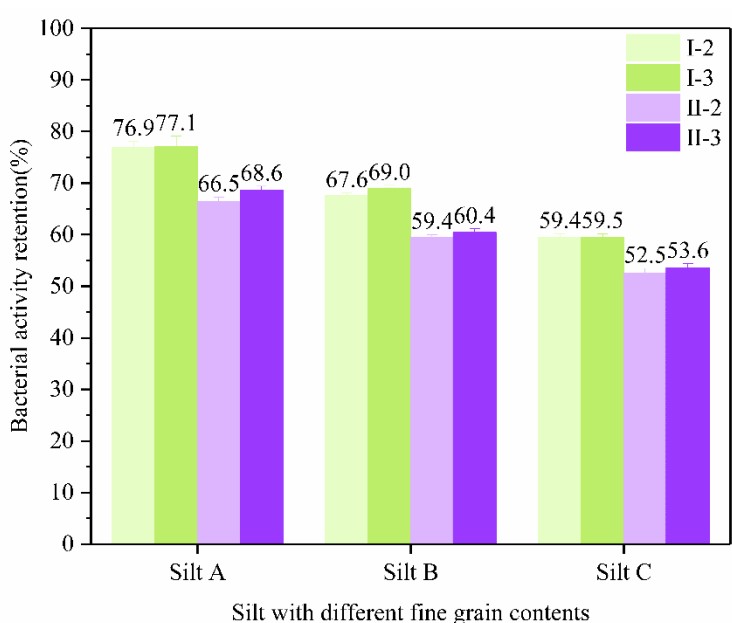

**Figure 9.** Effect of different fine particle content on the ability to retain bacteria.

### 3.4. Relationship between the Ability of Retaining Bacteria and CaCO$_3$

As can be seen from Figure 10, after curing by microorganism, the CaCO$_3$ content of silt increased with the improvement of silt's ability to retain bacteria, while the ability of silt to retain bacteria increased with the decrease in porosity ratio and with the increase in fine particle content. In particular, the retention ability of silt A with pore ratio of 0.9 and fine particle content of 75% was better than that of silt B with pore ratio of 0.75 and fine particle content of 47%, indicating that the retention ability of silt bacteria was jointly affected by the pore ratio and the fine particle content. When the pore size was large, the fine particle content began to play a decisive role. On the whole, when all kinds of silt have

similar retention bacteria capacity, the percentage of CaCO$_3$ is higher when the injection amount of cement solution is 300 mL per round.

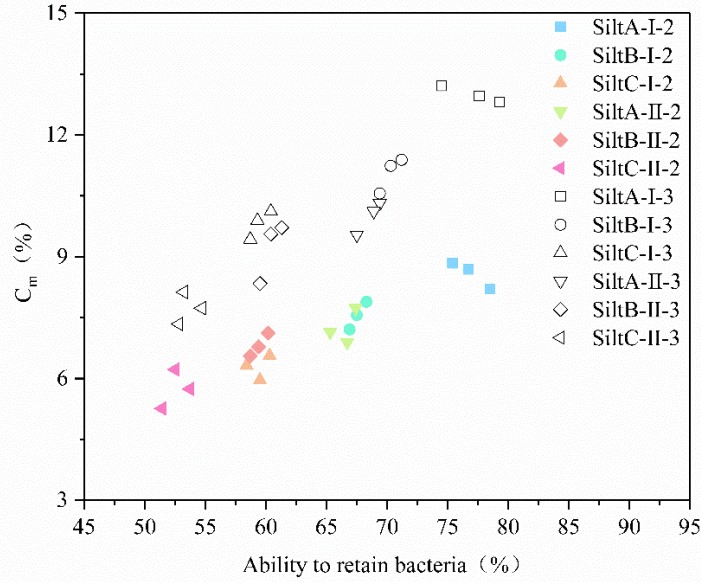

**Figure 10.** Relationship between retention bacteria capacity and CaCO$_3$.

### 3.5. Comparison of Uniaxial Compressive Strength

As can be seen from Figures 11 and 12, the unconfined compressive strength of silt after reinforcement is significantly improved compared with that before treatment. Gao et al. [17] reported that the shear strength of the biocemented samples was higher than that of untreated samples at all relative density levels. The more treatments, the higher the shear strength, and the biocemented samples showed greater initial stiffness than untreated samples. In the case of the same amount of bacterial fluid injection, the effect of cement fluid injection on silt strength is not linear. With the injection of cementing fluid, the pores between particles are filled with calcium carbonate, which leads to the decrease in the conversion rate of calcium carbonate, so the strength of silt will not be doubled.

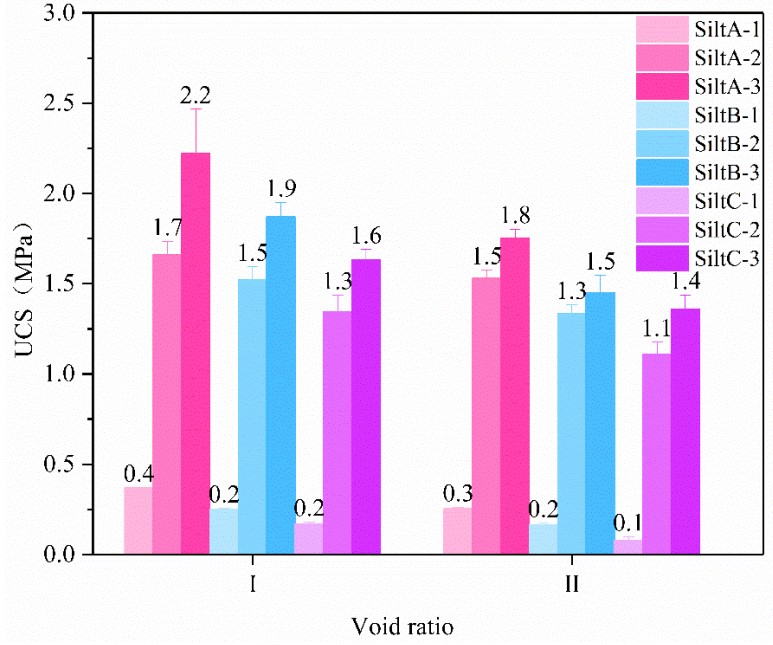

**Figure 11.** Influence of different pore ratios on unconfined compressive strength.

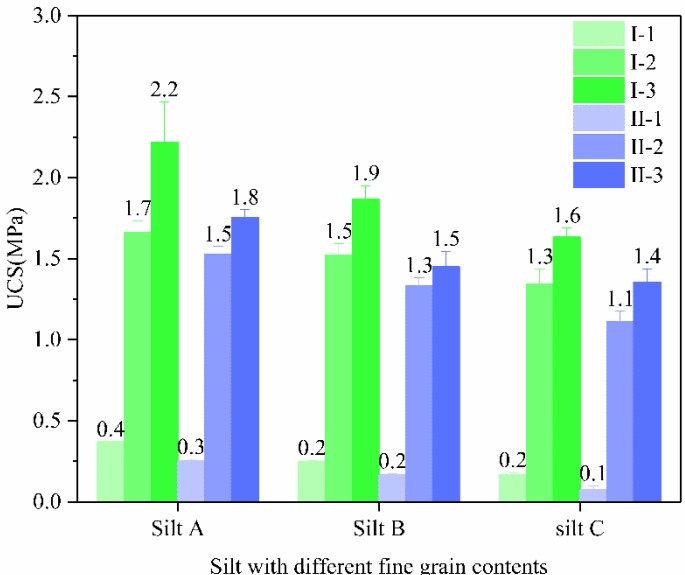

**Figure 12.** Effects of different fine particle content on unconfined compressive strength.

As can be seen from Figure 12, unconfined compressive strength increases with the increase in fine particle content under the condition of the same porosity ratio, and this rule can be reflected in both untreated silt and strengthened silt. When silt is untreated, the fine particles provide cohesive force for the particles to bond and thus have a certain strength. When the silt is treated through biochemical techniques, the fine particles will make more bacterial liquid and cementation liquid remain in the silt, and generate calcium carbonate on the particle surface and in the pores to cement the soil. Martinez and DeJong [27] reports that when silica sand is used, the degradation of cemented sand particles at the micro scale is controlled by calcite bonds. Therefore, it can be expected that denser soils with higher particle contact numbers will have larger increases in unconfined compressive strength than loose soils.

As can be seen from Figure 11, in the case of the same fine particle content, the unconfined compressive strength of silt with a pore ratio of 0.75 is slightly higher than that of silt with a pore ratio of 0.9 of the same type. When the pore ratio is 0.75, the pore volume of silt sample is small and large. When the microbe induced calcium carbonate precipitation technique was applied to the sample, the number of bacteria in the pores of the samples increased, which increased the nucleation site of calcium carbonate when the cementing solution was injected, and the content of calcium carbonate increased correspondingly and the strength increased. When the pore ratio is 0.9, the pore volume of the silt sample is too large, and the bacterial liquid and the cementing liquid are not easily retained in the sample during the grouting process, resulting in the reduction in calcium carbonate production and strength. By comparing the amount of calcium carbonate in sand and residual soil samples, Soon et al. [13] found that soil particle size had a significant effect on the amount of calcium carbonate. The calcium carbonate content of the treated sand samples is generally higher than that of the residual soil samples, but the improvement effect of microbial-induced calcium carbonate precipitation on the shear strength of the residual soil is better than that of the sand, and the increase in the shear strength of the residual soil increases with the increase in density.

### 3.6. Relationship between $CaCO_3$ and Unconfined Compressive Strength

As can be seen from Figure 13, unconfined compressive strength of the sample is significantly correlated with $CaCO_3$ content. With the increase in $CaCO_3$ content, the distribution of $CaCO_3$ in the sample is more uniform. When the content of $CaCO_3$ is accumulated to a certain extent, the strength of the sample will be significantly improved. Chu et al. [28] found that there was a strong linear correlation between unconfined compressive strength

of solidified sand column and calcite deposition, and the corresponding linear empirical formula was obtained.

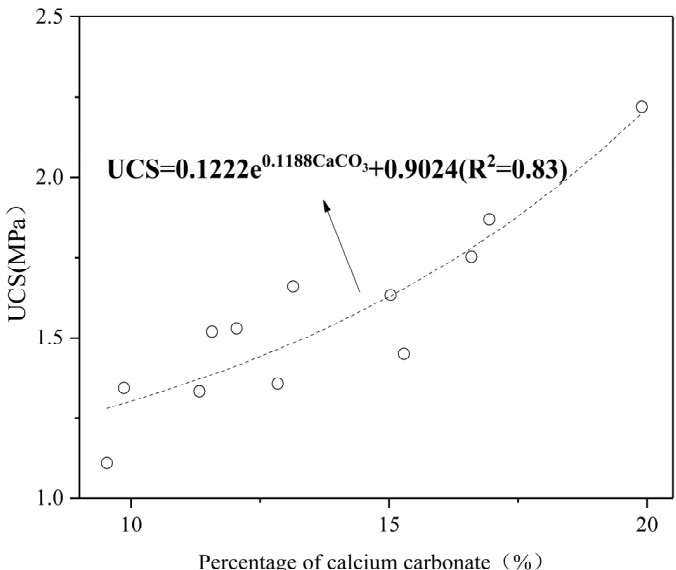

**Figure 13.** Relationship between the content of calcium carbonate and unconfined compressive strength.

### 3.7. Comparison of Permeability Coefficient

As can be seen from Figures 14 and 15, after curing by microorganism, the permeability coefficient of Silt A with pore ratio of 0.75 and fine particle content of 75% is $7.08 \times 10^{-8}$ cm/s, while that of the same type of untreated silt is $1.18 \times 10^{-4}$ cm/s, which decreases by 4 orders of magnitude. This indicates that calcium carbonate precipitation induced by microorganisms technology can significantly improve the permeability coefficient of silt, making it nearly impervious to water. It can be seen from the figure that, when the injection amount of single round cementing fluid is 150 mL, the permeability coefficient of silt has decreased significantly compared with that of untreated silt, and when the injection amount of cementing fluid continues to increase, the permeability coefficient can still decrease by 1~2 orders of magnitude.

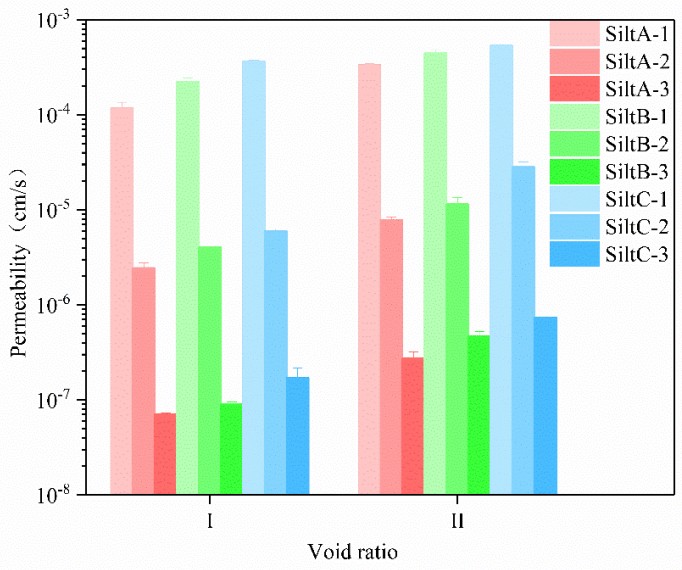

**Figure 14.** Influence of different pore ratio on permeability coefficient.

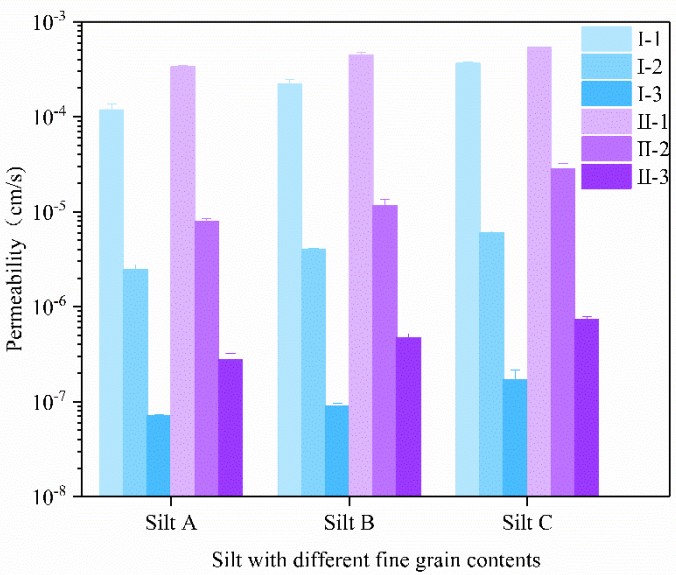

**Figure 15.** Effect of different fine particle content on permeability coefficient.

As can be seen from Figure 14, when the porosity ratio was 0.75 and the cement injection amount was 150 mL, the permeability coefficient of silt with different fine particle contents decreased by 2 orders of magnitude compared with that of untreated silt. When the pore ratio is 0.9, the permeability coefficient of silt with different fine particle content does not decrease significantly because of the large and abundant pores in the samples. On the contrary, the permeability coefficient of silt with fine particle content of 63% and 47% only decreases by one order of magnitude compared with that without treatment, which is similar to the production amount of $CaCO_3$. All are caused by the large porosity in the sample. Al Qabany et al. [29] conducted permeability tests on samples with different relative densities, and the results showed that with the increase in calcium carbonate precipitation amount, the permeability of both loose and compact samples decreased.

## 4. Conclusions

In this paper, based on calcium carbonate precipitation induced by microorganisms technology, considering the influence of three different factors, such as fine particle content, porosity ratio and cement injection amount, on the curing effect of silt, a series of tests, such as $CaCO_3$, unconfined compressive strength, permeability, XRD, SEM and so on, show that the macroscopic properties and microstructure of the strengthened silt samples have been improved to varying degrees.

1.  After curing by microorganism, the $CaCO_3$ content of silt increased with the decrease in pore ratio and the increase in fine particle content. Therefore, the $CaCO_3$ content of Silt A with pore ratio of 0.75 and fine particle content of 75% was the highest. There is no linear relationship between the content of $CaCO_3$ in silt and the injection amount of cementing liquid. When the injection amount of cementing liquid is 150 mL in a single round, the $CaCO_3$ content is 13%, and when the injection amount is 300 mL, the $CaCO_3$ content is 19%.
2.  By XRD analysis, there are three crystal types of calcium carbonate generated in the biological treatment samples, namely aragonite, vaterite and calcite, among which vaterite and aragonite are the main crystal types. The SEM images of three kinds of biological treatment samples of silt with different fine particle contents were obtained, and the crystal shapes were mainly spherical and acicular clusters, which were consistent with the XRD results. With the decrease in fine particle content, the crystal shape changes from spherical uniform distribution to spherical and acicular cluster distribution.

3. In the process of calcium carbonate precipitation induced by microorganisms, the capacity of residual bacteria in silt under different conditions is all over 50%. The higher the capacity of residual bacteria is, the higher the content of $CaCO_3$ in silt after curing. The retention capacity of silt A was affected by silt porosity ratio and fine particle content, and had nothing to do with the injection amount of cementing fluid. The retention capacity of silt A with pore ratio of 0.75 and fine particle content of 75% was the highest, which was 77%.

4. Untreated silt has a certain strength due to its cohesion. After curing by microorganism, the unconfined compressive strength of silt can reach more than 1 MPa. When the content of fine particles is the same, the unconfined compressive strength of silt with a pore ratio of 0.75 is higher than that of silt with a pore ratio of 0.9 because of its small pore volume, large number of pores and strong retention bacteria capacity. There is an obvious correlation between unconfined compressive strength and $CaCO_3$ content of silt. When $CaCO_3$ content accumulates to a certain extent, the strength of silt will be significantly improved.

5. The permeability coefficient of untreated silt is in the range of $1.18 \times 10^{-4} \sim 5.37 \times 10^{-4}$ cm/s. After curing by microorganism, the permeability coefficient of silt is significantly improved. In particular, when the single injection amount of cementing fluid is 300 mL, the permeability coefficient of Silt A with pore ratio of 0.75 and fine particle content of 75% is $7.08 \times 10^{-8}$ cm/s, and the silt is close to the impermeable state.

**Author Contributions:** Conceptualization, Z.X. and X.H.; Data curation, Q.W.; Formal analysis, Q.W. and Z.X.; Funding acquisition, Y.Z., J.Z. and Q.T.; Investigation, Y.Z.; Methodology, Y.Z. and Q.W.; Project administration, Q.T.; Resources, Y.Z.; Software, Q.W., M.Y. and X.C.; Supervision, Y.Z., X.H. and J.Z.; Validation, Q.T.; Writing—original draft preparation, Y.Z.; Writing—review and editing, Q.W. All authors have read and agreed to the published version of the manuscript.

**Funding:** The research was supported by National Natural Science Foundation of China (No.51409102), Postdoctoral Science Foundation of China (No. 2018M640683), Postdoctoral Research Grant of Henan (No. 001801006), Henan Science Foundation (No. 19A560003; 202102310011), Key Research Projects of Henan Higher Education Institutions (No. 19A320045); National Nature Science Foundation of China (52078317); Natural Science Foundation of Jiangsu Province for Excellent Young Scholars (BK20211597); Bureau of Housing and Urban-Rural Development of Suzhou (2021–2025) and Youth talent teacher of Henan (No. 2018GGJS078).

**Institutional Review Board Statement:** Not applicable.

**Informed Consent Statement:** Not applicable.

**Data Availability Statement:** Not applicable.

**Acknowledgments:** The Map of China quoted in this paper comes from the Standard Map Service and is produced under the supervision of the Ministry of Natural Resources. The map number is GS(2016)1569.

**Conflicts of Interest:** The authors declare no conflict of interest.

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
