# Peer review of "The Effect of MICP on Physical and Mechanical Properties of Silt with Different Fine Particle Content and Pore Ratio"

_applsci, doi:10.3390/app12010139_

Round 1

Reviewer 1 Report

This paper entitled “The effect of MICP on physical and mechanical properties of silt with different fine particle content and pore ratio” . To date, most of the MICP research has focused on treating various kinds of sands, Few studies have been done on other soil type. In this paper, the effects of different fine particle content and pore ratio on the physical and chemical properties of the reinforced silt were studied based on microbial-induced calcium carbonate technology.This is also an innovation of this paper. On the whole, the structure of the paper is rigorous, the experimental data is perfect and the expression is perfect. However, there are also some problems. The specific opinions are as follows: 1.Avoid jargon, such as "bio-treated" or "bio-cemented". Instead, use neutral wording, e.g. "cemented in the presence of bacteria", etc. 2. Avoid unnecessary acronyms (MICP, UCS, etc). This makes the text difficult to read for a broad audience and, ultimately, reduces the impact of the study. 3. In paragraph 1 of 2.2 Soil Properties and Gradation, "In order to obtain the soil with fine particle content between the two areas, the Yellow River soil and Jinan soil were mixed in a ratio of 3:7, And the fine particle content of the mixed silt was 63%." The Yellow River soil in the description scope is too large. 4.In paragraph 2 of 2.2 Soil Properties and Gradation, According to "For the convenience of description, Silt A, Silt B and Silt C are used to represent the three silts according to their fine particle contents from high to low, respectively”, thus we concluded that in ig.2 and Table1, the information of silt B and silt C should be exchanged. 5.What do the numbers 1, 2 and 3 in the figures mean. 6.The positions of calcium carbonate and silt should be marked in detail in Fig.7 SEM. Recommendation:this manu may be accepted after minor modification.

Reviewer 2 Report

It is recommended that most of the illustrations on grain size variation should be presented in color bar scale, as well as the other illustrations. Mineralogy does not use CaCO3 crystals, but CaCO3 polymorphs. Please carry the corrections. Also, correct the chemical formulas by considering the subscript numbers.

Please make a more comprehensive discussion, considering chemical-mineralogical and environmental processes. You could insert a thermodynamic stability graph for the three polymorphs to enrich the discussion.  
